# Chloramphenicol and gentamicin reduce the evolution of resistance to phage ΦX174 by suppressing a subset of *E. coli* LPS mutants

**Lavisha Parab**[1]*, **Jordan Romeyer Dherbey**[1], **Norma Rivera**[1], **Michael Schwarz**[1], **Jenna Gallie**[2], **Frederic Bertels**[1]*

1 Microbial Molecular Evolution Group, Department of Microbial Population Biology, Max Planck Institute for Evolutionary Biology, Plön, Germany, 2 Microbial Evolutionary Dynamics Group, Department of Theoretical Biology, Max Planck Institute for Evolutionary Biology, Plön, Germany

* parab@evolbio.mpg.de (LP); bertels@evolbio.mpg.de (FB)

**Data Availability Statement:** Antibiotic susceptibility. Agar plate images and raw greyscale values (all replicates) used to generate Fig 2 and S1

## Abstract

Bacteriophages infect gram-negative bacteria by attaching to molecules present on the bacterial surface, often lipopolysaccharides (LPS). Modification of LPS can lead to resistance to phage infection. In addition, LPS modifications can impact antibiotic susceptibility, allowing for phage–antibiotic synergism. The evolutionary mechanism(s) behind such synergistic interactions remain largely unclear. Here, we show that the presence of antibiotics can affect the evolution of resistance to phage infection, using phage ΦX174 and *Escherichia coli* C. We use a collection of 34 *E. coli* C LPS strains, each of which is resistant to ΦX174, and has either a "rough" or "deep rough" LPS phenotype. Growth of the bacterial strains with the deep rough phenotype is inhibited at low concentrations of chloramphenicol and, to a much lesser degree, gentamicin. Treating *E. coli* C wild type with ΦX174 and chloramphenicol eliminates the emergence of mutants with the deep rough phenotype, and thereby slows the evolution of resistance to phage infection. At slightly lower chloramphenicol concentrations, phage resistance rates are similar to those observed at high concentrations; yet, we show that the diversity of possible mutants is much larger than at higher chloramphenicol concentrations. These data suggest that specific antibiotic concentrations can lead to synergistic phage–antibiotic interactions that disappear at higher antibiotic concentrations. Overall, we show that the change in survival of various ΦX174-resistant *E. coli* C mutants in the presence of antibiotics can explain the observed phage–antibiotic synergism.

## Introduction

The rise of antibiotic-resistant bacteria is a global threat to public health [1]. To tackle this issue, several alternatives to antibiotics have been proposed [2]. Among the most promising of these are bacteriophages (phages), which have been successfully applied to a growing number of antibiotic-resistant bacterial infections [3–6]. However, as with antibiotics, a major reason for failure of phage treatments is the evolution of resistance [7]. The success of phages as therapeutic agents depends on our ability to limit the emergence of phage-resistant bacteria.

Table are available in Edmond (https://doi.org/10.17617/3.PIDVUT). Whole genome re-sequencing. The *E. coli* C wildtype genome file (used as a reference), the ΦX174 wildtype genome (used as a reference), and raw sequencing reads of all sequenced phage-resistant *E. coli* C isolates are available in Edmond (https://doi.org/10.17617/3.PIDVUT). Bootstrap of fluctuation experiment. Raw bootstrap values used to plot error bars in Fig 5 and Fig S5 in S2 Text can be found in Edmond (https://doi.org/10.17617/3.PIDVUT) under file Figure5_bootstrap_observedmutations.xlsx. The code used for bootstrapping iterations and chi-squared test (Table 2) is available as chi_squared_test_models.R, and the input file for the code is available as chi_squared_test.xlsx in Edmond (https://doi.org/10.17617/3.PIDVUT). Two-day coevolution experiment. The 96-well plate-layout, growth curves for individual replicates and raw reads (growth data, source for Fig 6) for both days are available in Edmond (https://doi.org/10.17617/3.PIDVUT).

**Funding:** This work was generously supported by funds from the Max Planck Society (L.P.-F.B.). L.P. was supported by the International Max Planck Research School for Evolutionary Biology (IMPRS EvolBio). The funders had no role in study design, data collection and analysis, decision to publish, or preparation of the manuscript.

**Competing interests:** The authors have declared that no competing interests exist.

**Abbreviations:** HSD, Honest Significant Difference; LB, lysogeny broth; LPS, lipopolysaccharide; LRT, likelihood-ratio test; MOI, multiplicity of infection.

### Reducing the evolution of phage-resistance in bacteria

One way to reduce the evolution of resistance is co-treatment of infections with specific combinations of phages and antibiotics [8–12]. Under these conditions, bacteria must evolve resistance to both the phage and the antibiotic to evade elimination. Interestingly, however, the efficacy of combined treatments is not always intuitive: examples of both synergistic and antagonistic interactions have been described. For example, phages NP1 and NP3 display synergistic effects with ceftazidime and ciprofloxacin when killing *Pseudomonas aeruginosa* biofilms [10]. Phages combined with antibiotics led to between a hundred-fold and thousand-fold higher killing rates, compared with phages alone. Contrastingly, phage SBW25Φ2 combined with streptomycin has been shown to increase rates of phage-resistance in *Pseudomonas fluorescens* populations [13]. While it is clear that the presence of antibiotics can affect the evolution of phage-resistance, thus far the underlying mechanisms remain largely unexplored. In this work, we provide a detailed example of how, and why, antibiotics can affect the course of phage-resistance evolution in bacterial populations.

### Mechanisms of phage-resistance in bacteria

A common mechanism by which bacteria become resistant to phage is through the modification of cell surface components, such as lipopolysaccharide (LPS) molecules [14–16]. LPS molecules are found on the outer membrane of all gram-negative bacteria and are the most common phage receptor of gram-negative-infecting phages [17,18]. Most *Escherichia coli* strains produce a "smooth" LPS phenotype, consisting of a lipid A domain, a core oligosaccharide (consisting of an inner-core and an outer-core), and a polysaccharide called the O-antigen [19]. Some strains, such as *E. coli* C, lack the O-antigen and hence produce a "rough" LPS phenotype [20]. Further truncation of the LPS by removal of the outer-core oligosaccharide leads to a "deep rough" LPS phenotype [21].

### *E. coli* C readily evolves resistance to ΦX174 through LPS alterations

Phage ΦX174 uses LPS to infect *E. coli* C [22]. ΦX174 is one of the oldest and most widely used phage model systems [23–25]. Surprisingly though, the mechanisms by which *E. coli* C can gain resistance to ΦX174—and how these can be overcome—have only been explored recently [16]. In this previous study, we showed that *E. coli* C populations readily become resistant to ΦX174 infection through mutations that generate truncated LPS structures (**Fig 1**) and that ΦX174 can evolve to overcome this type of resistance.

### LPS phenotype affects sensitivity to both phage and antibiotics

In addition to affecting infection by phages, LPS phenotype can influence susceptibility to antibiotics. In *E. coli*, deep rough LPS mutants are more susceptible to membrane-targeting antibiotics such as colistin [26,27]. Resistance to many antibiotics that do not target the membrane can also be affected, because LPS modifications can also change membrane permeability [21,28]. For example, *E. coli* K-12 strains carrying a deep rough LPS structure are more susceptible to hydrophobic molecules such as novobiocin and spiramycin [21,28]. However, truncated LPS molecules can also lead to increased antibiotic-resistance. In *P. aeruginosa*, rough mutants (carrying mutations in *galU*) can increase antibiotic-resistance [29,30]. Hence, only some combinations of LPS-targeting phages, antibiotic, and target pathogen, are expected to reduce the evolution of resistance.

In the current study, we investigate the effects of antibiotics on the evolution of ΦX174-resistance in *E. coli* C. We begin by measuring the susceptibility of a set of 33 ΦX174-resistant *E. coli* C

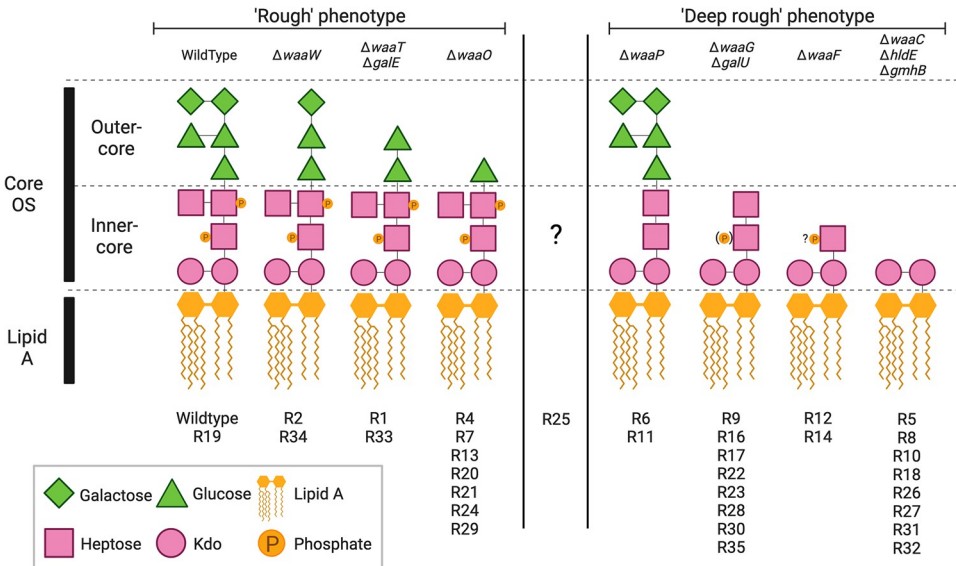

**Fig 1. Predicted LPS phenotypes of 33 phage-resistant *E. coli* C strains.** Strains (R#) are categorised into "rough" (longer; outer-core present but modified, inner-core unmodified) or "deep-rough" (shorter; outer-core absent and/or inner-core modified), based on mutation location. LPS phenotype cannot be predicted for R25 as the mutated gene, *rfaH*, controls the expression of the entire *waa* LPS operon [31]. While the presence or absence of LPS outer core is commonly used to categorise LPS phenotypes, previous studies with gene deletion mutants show that absence of phosphate molecules in the inner-core also leads to a "deep rough" phenotype, as seen in R6 and R11 [20]. Kdo: 3-deoxy-D-*manno*-octulosonic acid; "?P": No information was found on the phosphorylation of Hep(I) in the absence of *waaF*; (P): only 40% of the hexose phosphorylation was observed [32]. OS: oligosaccharide. Figure adapted from [16] with BioRender.com.

strains [16] to chloramphenicol and gentamicin. We find that *E. coli* C mutants with progressively shorter LPS molecules tend to be increasingly susceptible to both antibiotics (especially chloramphenicol). These results suggest that phage–antibiotic combination treatment prevents the growth of some antibiotic-resistant bacteria, and hence reduces the number and diversity of observable resistance mutations. The results of subsequent fluctuation experiments are consistent with this prediction; no *E. coli* C mutants with severely truncated LPS molecules were detected in a ΦX174 +chloramphenicol combination environment. Finally, we use an *E. coli* C and ΦX174 co-evolution experiment in liquid media to show that, indeed, the ΦX174+chloramphenicol combination environment is the most efficient (of all combination treatments tested) at reducing phage-resistance evolution. Together, our results show that predicting phage-resistance rates in bacteria can be possible with surprising accuracy, using only very simple susceptibility assays.

## Results

### Phage-resistance mutations affect *E. coli* C susceptibility to chloramphenicol and gentamicin

The effect of LPS phenotype on antibiotic susceptibility is expected to depend on the antibiotic in question. Hence, in our first experiment, we investigated the effect of LPS phenotype on the susceptibility of *E. coli* C to two different antibiotics: chloramphenicol and gentamicin. These antibiotics inhibit protein synthesis via distinct mechanisms, respectively targeting the 50S and 30S ribosomal subunits [33]. We quantified the growth of 34 *E. coli* C strains with various predicted LPS phenotypes (see **Fig 1**) in the presence of increasing—but nonetheless low— concentrations of chloramphenicol or gentamicin on agar plates. The results are presented in **Fig 2** and discussed in more detail below.

**Chloramphenicol.** The degree to which chloramphenicol affects *E. coli* C growth depends on LPS phenotype: progressively truncated LPS structures lead to increasing chloramphenicol susceptibility (**Fig 2A** and **Fig S1** in **S2 Text**). More specifically, *E. coli* C strains displaying the rough LPS phenotype (including the wild type) showed no detectable growth at chloramphenicol concentrations of >3–4 μg/ml, while the growth of strains displaying the deep rough LPS phenotype was already inhibited at chloramphenicol concentrations of 2 μg/ml or less (**Fig 2B** and **Fig S3** in **S2 Text**). Further, a subset of deep rough mutants predicted to possess the shortest LPS structures (R8, R10, R18, and R26) were most susceptible to chloramphenicol, with growth inhibition occurring at 1 μg/ml or lower.

**Gentamicin.** Compared with chloramphenicol, susceptibility to gentamicin is less dependent on LPS phenotype. Many of the phage-resistant strains showed a similar growth profile to that of the wild type across the gentamicin concentrations tested, with significant growth inhibition first occurring at concentrations of 3 to 4 μg/ml (**Fig 2A and 2B** and **Figs S2 and S3** in **S2 Text**). One exception was a subset of 6 strains with severely truncated LPS phenotypes (*hldE*/*waaC* mutants: R8, R10, R18, R26, R31, R32), the growth of which was inhibited at lower levels of gentamicin (1.5 to 2 μg/ml) (see also **Note A in S1 Text**). Interestingly, the growth of 5 strains—namely, 4 strain with the rough LPS phenotype (R34, R1, R33, R20) plus the strain of unknown LPS phenotype (R25)—was first inhibited at a slightly higher gentamicin concentration than the wild type (6 μg/ml versus 4 μg/ml). Hence, one might expect similar LPS mutations to be observed in gentamicin resistance screens. This is indeed the case for low-to-moderate levels of gentamicin [29,34].

The results in this section demonstrate that LPS phenotypes—and hence, phage-resistance genotypes—affect the susceptibility of *E. coli* C to both chloramphenicol and gentamicin. In general terms, shorter LPS structures tend to increase susceptibility to the antibiotic particularly chloramphenicol.

## The presence of antibiotics is predicted to reduce diversity of phage-resistance mutations

In the previous section, we demonstrated that LPS mutations that decrease the susceptibility of *E. coli* C to ΦX174 can simultaneously increase susceptibility to low levels of antibiotics. This relationship suggests that the presence of antibiotics may affect the evolution of phage-resistance in *E. coli* C populations (and vice versa). Specifically, the frequency of phage-resistance mutations that render the bacterium more susceptible to antibiotic (i.e., those conferring the deep rough LPS phenotype) is expected to reduce in the presence of the antibiotic. In this section, we build a simple model to demonstrate this idea (model 1). This model can be outlined as follows:

1. Phage-resistance is primarily caused by deactivating mutation(s) in gene(s) encoding the LPS biosynthetic machinery [16].

2. If we assume that these deactivating mutations occur uniformly between, and across the length of, each LPS gene, the total length of the gene(s) giving rise to phage resistance (i.e., the mutational target size) can be inferred.

3. Due to difference in which LPS phenotypes are permissible the mutational target size is affected by the presence—and concentration of—antibiotic. This information can be used to predict the relative rates of phage-resistance across environments.

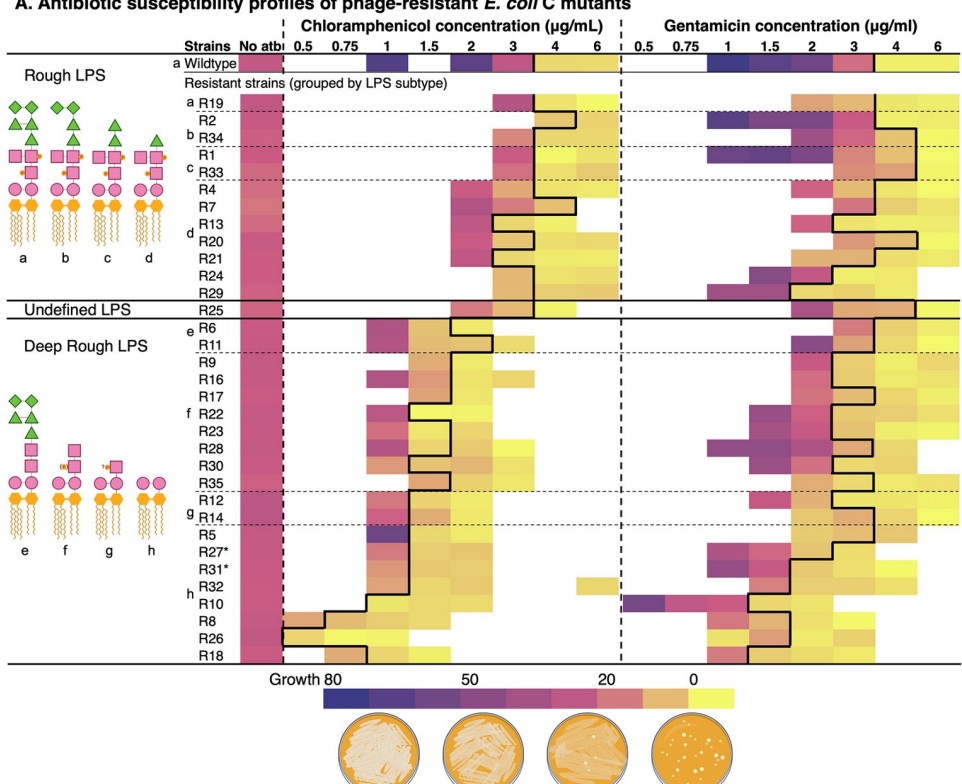

**A. Antibiotic susceptibility profiles of phage-resistant *E. coli* C mutants**

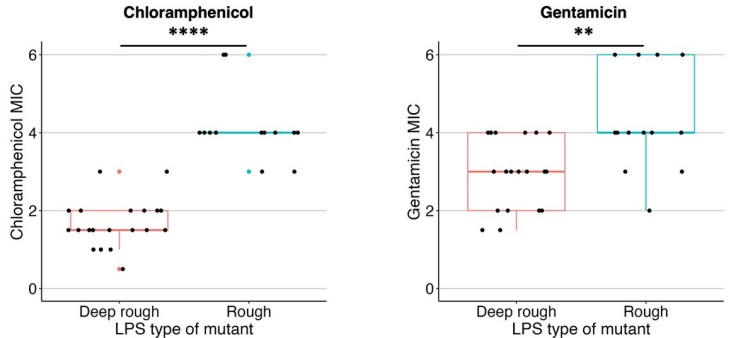

**B. Antibiotic minimum inhibitory concentration of phage-resistant mutants, separated by LPS types**

**Fig 2. *E. coli* C deep rough LPS mutants show increased susceptibility to chloramphenicol and gentamicin.**
Heatmaps show growth of *E. coli* C wild type and 33 *E. coli* C-derived, phage-resistant strains (R#) at increasing
concentrations of chloramphenicol (left) and gentamicin (right). Growth is measured in 8-bit pixel greyscale values
(see **Methods**) and decreases from violet (uniform lawn = greyscale values of 30 to 80) to yellow (no lawn = 0 to 8, and
occasional colonies may be visible). White = not measured. Median greyscale values above 8 are defined as "growth"
(left of bold black border), while values below 8 are defined as "no growth" (see **Methods**). Bacterial strains (rows) are
ordered according to their predicted LPS phenotype (top to bottom: least to most truncated) and grouped as rough,
undefined, or deep rough. Dotted horizontal lines group strains that have the same predicted LPS structure (shown on
the left and indicated by letters a–h). *R27 and R31 have the same genotype (see **Note A in S1 Text**). The MIC of *E.
coli* C wild type is approximately 4 μg/ml for both antibiotics. Increased bacterial growth at sub-MIC antibiotic
concentrations is observed in some cases; similar observations have been described previously [35–38]. (**B**) Boxplots
show the MIC of the phage-resistant *E. coli* C strains in chloramphenicol (left) and gentamicin (right). The average
MIC of strains predicted to carry rough (longer) and deep rough (shorter) LPS molecules differs for both antibiotics (*t*
test *p* = 4.15E-08 and 0.002, respectively; Wilcoxon rank sum test for tied data sets *p* = 1.05E-08 and 0.005,
respectively). Bacterial strains predicted to display rough LPS are, on average, more susceptible to chloramphenicol
and gentamicin. The data underlying this figure can be found in https://doi.org/10.17617/3.PIDVUT which includes
agar plate images and greyscale values for each replicate (Batch_ROI_Grayscale_values.xlsx). Greyscale values are also
plotted in **Figs S1-S3 in S2 Text**; median values used in panel A are available in **S1 Table**. Clipart for panel A created
with BioRender.com.

**Table 1. A priori prediction of relative phage-resistance rates in various environments (model 1).**

| | Gene | | Mutant survival, if mutation in specified gene$ | | | | Mutation probability weights (proportional to gene length) | | | | |
|---|---|---|---|---|---|---|---|---|---|---|---|
| | Name | Length (bp) | CL 1 µg/ml | CL 2 µg/ml | GM 1 µg/ml | GM 2 µg/ml | Φ | Φ + CL 1 µg/ml | Φ + CL 2 µg/ml | Φ + GM 1 µg/ml | Φ + GM 2 µg/ml |
| R O U G H | *yajC*§ | 333 | Y* | Y | Y | Y | 333 | 333 | 333 | 333 | 333 |
| | *waaW* | 1,026 | Y | Y | Y | Y | 1,026 | 1,026 | 1,026 | 1,026 | 1,026 |
| | *galE* | 1,017 | Y | Y | Y | Y | 1,017 | 1,017 | 1,017 | 1,017 | 1,017 |
| | *waaT* | 996 | Y | Y | Y | Y | 996 | 996 | 996 | 996 | 996 |
| | *waaO* | 1,017 | Y | Y | Y | M (6/7) | 1,017 | 1,017 | 1,017 | 1,017 | 872 |
| | *rfaH* | 489 | Y | Y | Y | Y | 489 | 489 | 489 | 489 | 489 |
| D E E P R O U G H | *waaP* | 798 | Y | M (1/2) | Y | Y | 798 | 798 | 399 | 798 | 798 |
| | *waaG* | 1,125 | Y | N | Y | Y | 1,125 | 1,125 | 0 | 1,125 | 1,125 |
| | *galU* | 909 | Y | M (1/5) | Y | Y | 909 | 909 | 182 | 909 | 909 |
| | *waaF* | 1,047 | Y | N | Y | Y | 1,047 | 1,047 | 0 | 1,047 | 1,047 |
| | *gmhB* | 573 | Y | N | Y | Y | 573 | 573 | 0 | 573 | 573 |
| | *hldE* | 1,434 | M (3/6) | N | Y | M (1/6) | 1,434 | 717 | 0 | 1,434 | 244 |
| | *waaC* | 972 | N | N | Y | N | 972 | 0 | 0 | 972 | 0 |
| | Totals | 11,736 | | | | | 11,736 | 10,047 | 5,459 | 11,736 | 9,428 |
| | | | | | | Relative resistance rate# | 1.00 | 0.86 | 0.47 | 1.00 | 0.80 |

$Based on antibiotic susceptibilities of phage-resistant *E. coli* C strains from **Fig 2**.

*"Y" indicates all strains carrying a mutation in the specified gene are able to grow at the specified antibiotic concentration. "M (x/n)" indicates an intermediate phenotype: (x = number of strains that grow, *n* = total number of strains carrying a mutation in specified gene). "*N*" indicates none of the strains carrying a mutation in the specified gene grow. The model assumes that mutations occur with a probability proportional to the length of the gene in question. Thus, in genes with a "Y," the mutational target size is the coding region length. For "M (x/n)," the mutational target size is (x/n) multiplied by (gene length). For "N," the mutational target size is zero.

#The predicted relative resistance rate is the ratio of the mutational target size of the treatment to that of ΦX174 alone. CL: chloramphenicol, GM: gentamicin.

$*YajC* is the only gene detected that is not directly linked with LPS assembly or synthesis.

**Inferring permissible mutational target sizes.** The mutational target size can be inferred from the genes that contain loss-of-function mutations in the 33 phage-resistant strains [16]. Mutations in a total of 14 genes, with a combined length of 11,736 bp, led to phage-resistant mutants (**Table 1**). This mutational target size is specific to the ΦX174-only environment; as the antibiotic concentration increases, the number of genes in which phage-resistance mutations are accessible (i.e., lead to survival) decreases (see **Fig 2**). Such "inaccessible" genes can be deducted from the permissible target size in order to estimate the frequency of resistance to phage in each environment (**Table 1**).

The predicted relative resistance rates are provided in the final row of **Table 1** (and graphically in **Fig 3B**, dark red points). Overall, as antibiotic levels increase, the relative rate of phage-resistance is predicted to decrease. Most notably, the phage-resistance rate in 2 µg/ml chloramphenicol is predicted to be less than half (47%) of that expected when treating the bacteria only with ΦX174. Combination treatment with gentamicin reduces the predicted resistance by only 20% at the highest concentration tested (2 µg/ml).

## Observed phage-resistance rates are qualitatively in line with a priori predictions

In order to test the relative phage-resistance rates predicted in the a priori model above, a classic fluctuation experiment was performed [39,40]. A total of 430 independent *E. coli* C wild-type populations were grown (without phage or antibiotics). Each grown population was

## A. Fluctuation experiment setup

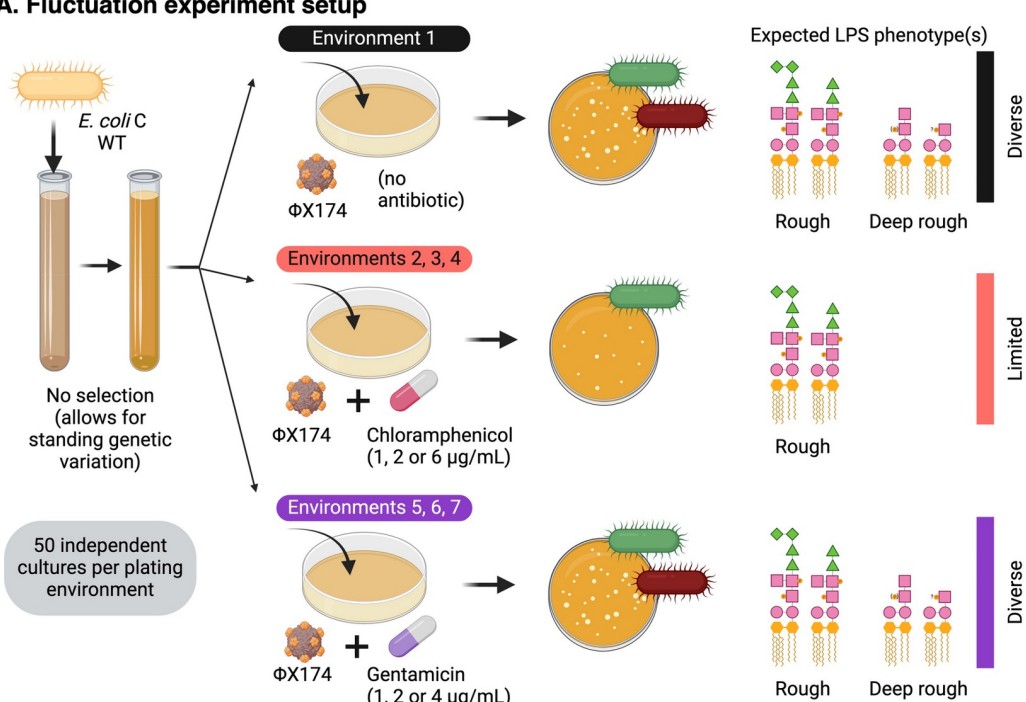

## B. Relative rates of phage-resistance (observed and predicted) in different environments

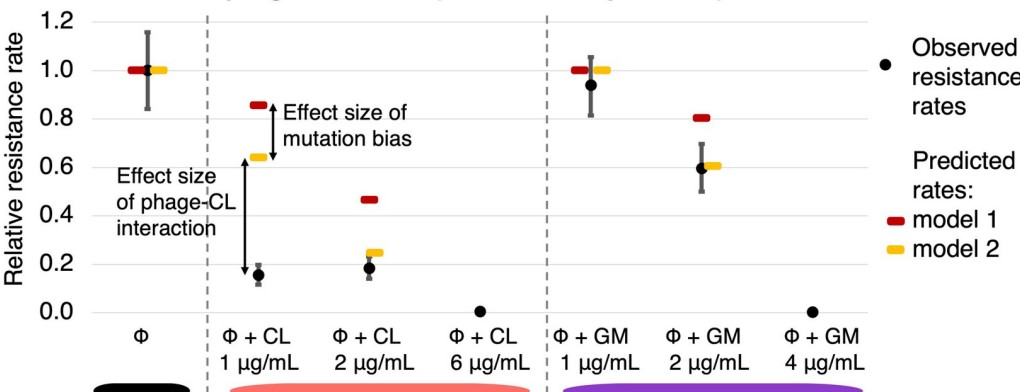

## C. Absolute rates of phage-resistance (observed) in different environments

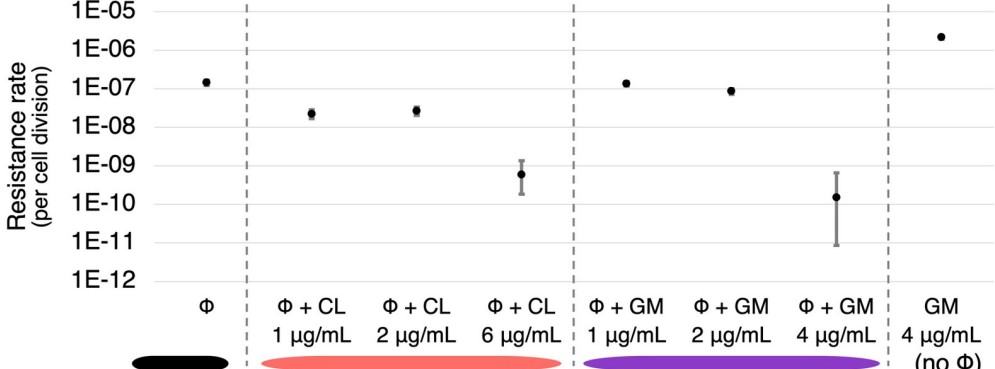

**Fig 3. Antibiotics reduce the rate of resistance to ΦX174. (A)** Schematic of the fluctuation test and expected results. Adding chloramphenicol at subinhibitory concentrations is predicted to reduce rates of resistance to ΦX174, due to inhibition of the growth of deep rough LPS phenotypes. Gentamicin, however, is predicted to have a much lower effect on the rate of resistance to ΦX174. **(B)** Resistance rates (and 95% confidence intervals) for each phage+antibiotic plating environment, relative to the resistance rate in the ΦX174-only environment. Mean observed rates (black) are qualitatively in line with the predicted rates from model 1 (gene length model, dark red; derived in **Table 1**). Rates predicted by model 2 (gene proportion model inferred from the occurrence of mutations in a phage-only fluctuation test (see below), yellow; derived in **S6 Table**) are within 95% confidence intervals for gentamicin (GM), and 2% away from the 95% confidence interval in 2 μg/ml chloramphenicol (CL). Resistance rates are not predicted at antibiotic concentrations inhibitory for wild type (6 μg/ml CL, 4 μg/ml GM). The difference between models 1 and 2 can be mostly attributed to mutation bias (see text). The difference between model 2 and the observed resistance rate is most likely due to the effect size of phage–antibiotic interactions. **(C)** Absolute resistance rates (per cell division and 95% confidence intervals), including gentamicin at wild-type MIC without ΦX174. Rates of ΦX174 with antibiotic at inhibitory concentrations seem to be zero in panel B but are actually ~400–1,000-fold lower, as can be seen in panel C. In panel C, some 95% confidence intervals are too small to be readily visible. The data underlying this figure can be found in **S2 Table** (resistance rates and other experimental values used for calculating them; see **Methods** for details) and **S3 Table** (raw colony counts). Panel A created with BioRender.com.

spread onto LB agar plates containing either (set 1) ΦX174 alone, (set 2 to 4) ΦX174 in combination with chloramphenicol (at 1 μg/ml, 2 μg/ml, or 6 μg/ml), (set 5 to 7) ΦX174 in combination with gentamicin (at 1 μg/ml, 2 μg/ml, or 4 μg/ml), a total of 50 replicates for each of the 7 environments (**Fig 3A**). In addition, 6 μg/ml chloramphenicol (set 8, *n* = 30) and 4 μg/ml gentamicin (set 9, *n* = 50) were used as control environments. For each plating environment, the number of resistant bacterial colonies arising from each population was counted. The distributions of colony numbers were used to calculate the observed relative phage-resistance rates [41] (**Fig 3B and 3C**).

In model 1, the addition of chloramphenicol was predicted to reduce the rate of phage-resistance in *E. coli* C (**Table 1**). Indeed, the addition of chloramphenicol did significantly reduce relative phage-resistance rates (likelihood-ratio test (LRT) $p < 10^{-5}$ for both 1 μg/ml and 2 μg/ml) (**S2 Table**). However, the observed relative rates of phage-resistance were far lower than expected: 15% at 1 μg/ml (~4-fold lower than the predicted 86%) and 18% (~2-fold lower than the predicted 47%) (**Fig 3B**). The addition of gentamicin was predicted, and observed, to have no effect on phage resistance rates at 1 μg/ml gentamicin (LRT $p$ = ~0.53). At 2 μg/ml gentamicin, the relative phage-resistance rate was reduced to 60%, somewhat further than was predicted (80%) (**Table 1**).

Overall, the addition of low concentrations of chloramphenicol or gentamicin did reduce the rate of phage-resistance in *E. coli* C (as predicted by model 1). However, the presence of antibiotics reduced the rate of phage-resistance much more effectively than predicted by the model. This could indicate that the reduction in rates of phage-resistance is not solely attributable to smaller mutational target sizes and that other factors should be considered for more accurate predictions. However, before considering additional factors, it is important to rule out the possibility that minor improvements to the existing model may lead to a better match with the observations.

## Chloramphenicol skews mutations towards genes causing a rough LPS phenotype

So far, the results have demonstrated that combining ΦX174 with chloramphenicol and, to a lesser degree, gentamicin reduces the rate of phage-resistance in *E. coli* C. In model 1, we hypothesised that this is—at least partly—due to a reduction in the number of accessible mutations that lead to phage-resistance. To directly test this hypothesis, 222 phage-resistant *E. coli* C mutants were isolated from five of the fluctuation test plating environments, and each subjected to whole genome re-sequencing. A total of 239 mutations, belonging to a range of

## A. Observed genomic distribution of phage-resistance mutations (in all environments)

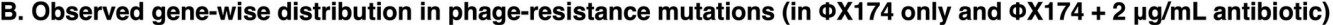

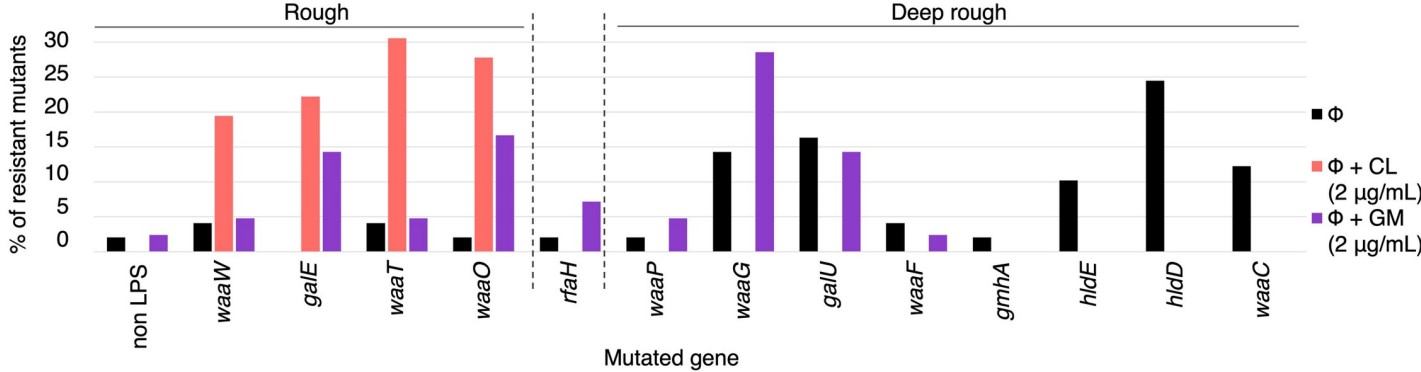

## B. Observed gene-wise distribution in phage-resistance mutations (in ΦX174 only and ΦX174 + 2 µg/mL antibiotic)

**Fig 4. The presence of antibiotics reduces mutational target size for phage-resistance evolution in *E. coli* C.** (A) Genomic distribution of phage-resistance mutations in *E. coli* C, under 5 plating environments of the fluctuation test (rows). Large deletions (>100 bp) are indicated by solid horizontal lines scaled to the deleted region; all other classes of mutations are indicated as dashes. Red boxes indicate loci in which mutations are absent (or extremely rare) under some environments. Genes contributing to LPS inner core synthesis are underlined and in red; loss-of-function mutations in these genes are predicted to give a deep rough LPS phenotype. *RfaH is responsible for regulation of the LPS operon, and *rfaH* mutations can lead to a mixture of rough and deep rough types [31,42]. (B) Gene-wise distribution of the phage-resistance mutations observed in the absence of, and presence of 2 µg/ml, antibiotics. Bars show the proportion of phage-resistant *E. coli* C mutants with a mutation in the specified gene (ΦX174-only *n* = 49, ΦX174+CL(1 µg/ml) *n* = 39, ΦX174+CL(2 µg/ml) *n* = 36, ΦX174+GM(1 µg/ml) *n* = 46, ΦX174+GM(2 µg/ml) *n* = 42). CL: chloramphenicol, GM: gentamicin. The raw data underlying this figure can be found in S4 Table.

mutational classes, were identified (**Fig 4** and **Fig S4** in **S2 Text** and **S4 Table**). Of the 222 mutant strains, 194 were found to contain mutations in the 14 previously identified LPS genes [16], and two new LPS genes were identified as mutational targets (*hldD* and *gmhA*). Some degree of parallel evolution (identical mutations) was observed, both within and between plating environments (**Note B in S1 Text**). We note that, in four mutant strains, no putative mutations were identified (possibly due to low coverage of the LPS biosynthetic operon, see **S4 Table**).

**Combining phage and antibiotics biases mutations towards rough LPS genes (Fig 5A).** Mutations in strains resistant to phage treatment alone are predominantly found in deep rough LPS genes (approximately 86%), and only about approximately 12% are in rough LPS genes. As expected from our predictions, adding chloramphenicol enriches for rough mutants. At 1 μg/ml chloramphenicol, the proportion of rough mutants increases from ~12% to ~62%, and the proportion of deep rough mutants decreases from ~86% to ~36%. At a concentration of 2 μg/ml chloramphenicol, all mutants are rough mutants. This result was somewhat surprising, since we predicted to observe at least some deep rough mutants (~11%) (**Fig 5A**). Gentamicin treatment also leads to a smaller proportion of deep rough mutants, but as expected from our predictions, the effect is much smaller. At a concentration of 1 μg/ml gentamicin deep rough mutants remain largely stable at ~85% and ~15% of the mutants are rough. At 2 μg/ml gentamicin, the ratio becomes more skewed towards a rough LPS with about 50% deep rough mutants and ~43% rough mutants (**Fig 5A**).

## Adjusted model improves predictions and highlights potential phage–antibiotic interactions

Before attempting to predict outcomes of combination environments, our model should be able to accurately predict outcomes in the ΦX174-only environment. In the ΦX174-only environment, deep rough mutants were observed at significantly higher rates (86%) than predicted from the mutational target size (model 1, 58%) (**Fig 5A** and **S5 Table**). One possible explanation for this mismatch is that the distribution of mutations is not actually proportional to gene length, as model 1 assumes. For example, mutational bias could lead to unexpectedly high (or low) rates of particular mutations [43,44]. To investigate this possibility, we used the proportion of mutants observed in each locus, in the ΦX174-only environment of the fluctuation test (a pseudo-count of 1 was added to each gene, to avoid false negatives, see **Methods**) (**S6 Table**). The adjusted predicted proportions are shown in yellow in **Fig 5B** (model 2). Adjusting the baseline predictions should, in theory, improve the predictions for the combination environments.

Using model 2 quantitatively improves predicted resistance rates (see **Fig 3B**), especially for environments with 2 μg/ml antibiotic (and phage). The new predicted resistance rate for 2 μg/ml gentamicin (61%) is now within the 95% confidence interval and close to the measured resistance rate (60%, with 95% CI 50% to 70%), while the predicted resistance rate in the 2 μg/ml chloramphenicol environment (i.e., 25%) now lies almost within the 95% confidence interval (18%, with 95% CI of 14% to 23%). However, the model 2-predicted distribution of rough and deep rough mutants fits the observed distribution less well than the model 1 predictions do (**Fig 5A**). The only environment where model 2 improved predictions of LPS mutant distribution above those in model 1 is ΦX174+1 μg/ml gentamicin (the environment closest to the ΦX174-only environment, based on our initial susceptibility assay) (**Fig 5A**). Model 2 also allows us to more accurately predict the effect of phage–antibiotic interactions on resistance rates by more accurately accounting for mutation bias (as shown in **Fig 3B**). The only environment where phage–antibiotic interactions appear to play a significant role is in the ΦX174+1 μg/ml chloramphenicol environment.

## A. Phage resistance mutations in all environments, grouped by LPS phenotype

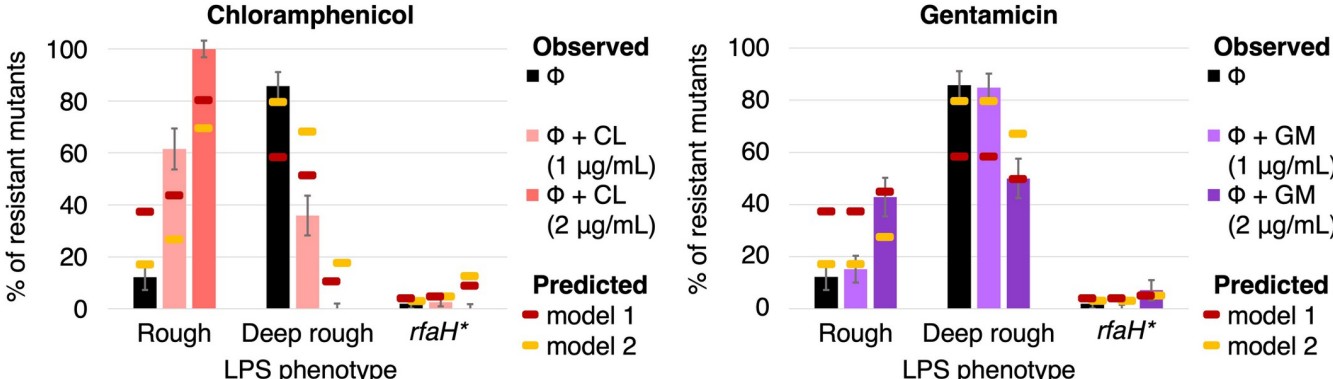

## B. Gene-wise phage resistance mutations in the ΦX174 only environment

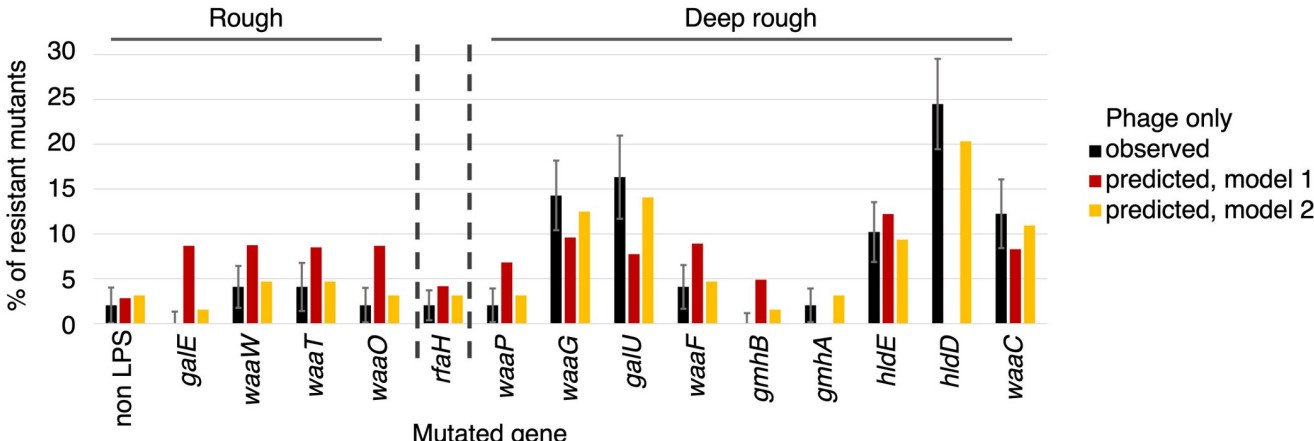

## C. Gene-wise phage resistance mutations in the ΦX174 + Chloramphenicol (2 µg/mL) environment

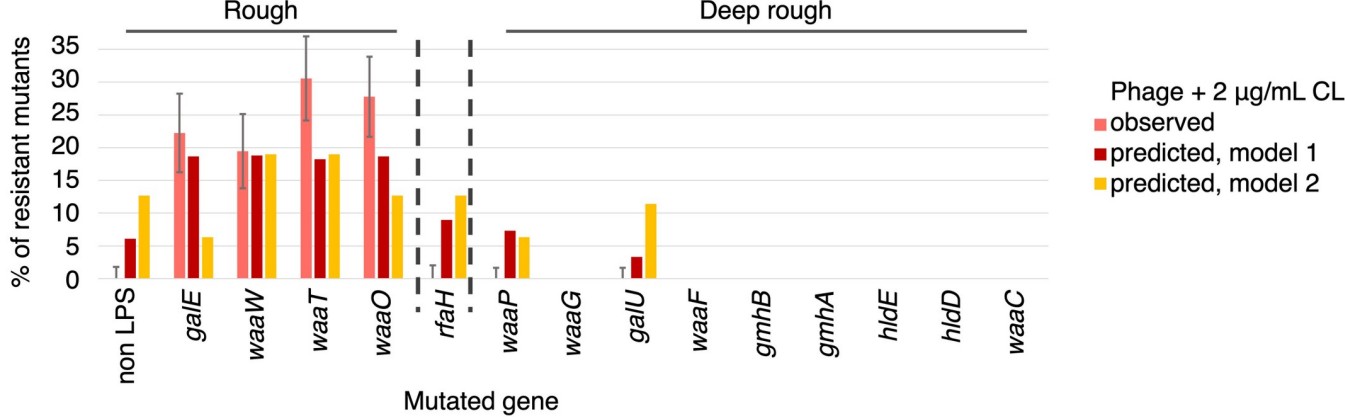

**Fig 5. Chloramphenicol heavily skews phage-resistance mutations towards rough LPS loci.** (**A**) Bars show the percentage of phage-resistant *E. coli* C mutants that carry mutations predicted to give rise to the rough or deep rough LPS phenotype, under various fluctuation test plating environments. Error bars show the standard deviation for each LPS type from 100 bootstrapping replicates (resampling with replacement, see **Methods**). Model 1 predictions (dark red points) assume that the probability of mutation is proportional to gene length (**Table 1**). In model 2 (dark yellow points), the probability of mutation is based on the observed proportions in the ΦX174-only environment (**S6 Table**). Predictions for ΦX174+antibiotic environments are based on the susceptibility data (see **Fig 2**). Mutations in *rfaH* can affect any number of inner or outer core genes, and hence are in a separate LPS phenotype category. (**B**) Gene-wise distribution of phage-resistance mutations, in the absence of any antibiotics. The proportions predicted by model 2 differ from the observed proportions in the ΦX174-only environment because we add a pseudo-count of 1 to counts for each of the genes, to allow statistical handling of zero values (see **Methods**). "Non LPS" refers to genes whose potential involvement in LPS biosynthesis, assembly, or regulation remains uncharacterized (4 genes across all experiments, see **Note C in S1 Text**). Error bars show the standard deviation from 100 bootstrap replicates for each gene (**B**,

C). (**C**) Gene-wise distribution of phage-resistance mutations, in the ΦX174+chloramphenicol (2 µg/ml) environment. As outlined in Model 2, *rfaH*, *galU*, and *waaP* mutants are predicted to survive, but none were observed. (See **Fig S5 in S2 Text** for model 3.) The data underlying this figure can be found in **S4**–**S7 Tables**, which include raw mutation data as well as summarised values. Raw bootstrapping values can be found in the file Figure5_bootstrap_observedmutations.xlsx in https://doi.org/10.17617/3.PIDVUT.

Model 2 qualitatively predicts the shift towards rough mutants, but without quantitative accuracy (**Fig 5A**). Since some *waaP* and *galU* mutants grow at 2 µg/ml chloramphenicol in our susceptibility assays, deep rough *waaP* and *galU* mutants are predicted to occur at a rate of 6% and 11%, leading to a non-zero deep rough prediction (**Fig 5A** and **5C and S6 Table**). *RfaH* mutants are predicted to occur at a rate of 13%, because they also grow well in our susceptibility assay. Yet, *waaP*, *galU*, and *rfaH* mutants were not observed at 2 µg/ml chloramphenicol (**Fig 5C**). Similarly, at 1 µg/ml chloramphenicol, mutations are predicted in *rfaH*, *waaF*, *gmhB*, and *gmhA* but none were observed (panel C of **Fig S5 in S2 Text**). At 2 µg/ml gentamicin, mutations are predicted in *hldE*, *gmhA*, and *gmhB* (deactivation of either is predicted to result in the deep rough LPS phenotype), but none were observed (panel D of **Fig S5 in S2 Text**).

There are at least three possible reasons for the above discrepancies between predictions and observations. First, sampling only 40 to 50 mutants per environment could under-sample the proportion of mutated LPS genes. Sampling more mutants in the ΦX174-only environment may improve the accuracy of predicted proportions (and prevent the need for the addition of pseudo-counts in our models). Second, biological interactions between antibiotics and phages [45,46] may lead to the overrepresentation of different LPS genes compared to the ΦX174-only environment. Third, it is possible that what is considered growth in our susceptibility assay is not sufficient to be detected in the fluctuation experiment.

We are planning to test hypotheses 1 and 2 in future studies, hypothesis 3 can be tested by adjusting the threshold for growth in our susceptibility assay. When we increase the threshold for growth from a greyscale value of 8 to 12 (model 3), phage-resistance rate predictions remain largely unchanged from those in model 2, with the exception of the 2 µg/ml gentamicin environment, where the prediction worsens (panel A of **Fig S5 in S2 Text**). Predicted distributions across LPS genes do not change significantly for most environments (panel B of **Fig S5 in S2 Text** and **S6**–**S7 Tables**), but model 3 does indeed improve the gene-wise predictions for phage+2 µg/ml antibiotic (panels D and E of **Fig S5 in S2 Text**). With the higher threshold, *gmhB*, *gmhA*, and *hldE mutants* are no longer predicted to grow at 2 µg/ml gentamicin (panel D of **Fig S5 in S2 Text**), neither are *waaP* and *galU* mutants at 2 µg/ml of chloramphenicol (panel E of **Fig S5 in S2 Text**), which matches our experimental observations. However, none of the model adjustments explain the discrepancy between the predicted and observed phage-resistance rates, or proportions of deep rough and rough mutants at 1 µg/ml chloramphenicol (panels A and B of **Fig S5 in S2 Text**). Hence, this discrepancy is most likely the result of direct or indirect interactions between phages and chloramphenicol.

Interactions between phages and chloramphenicol are also indicated by statistical comparisons between observations and expectations according to different models (**Table 2**). If there is no statistical difference between model and observation, we assume that the model performs well. As the difference between model and observation becomes more significant, the model fits the observations less well. Of the three models, models 2 and 3 provide the most accurate predictions of rough and deep rough mutant proportions (**Table 2**), and phage-resistance rates (**Fig 3B** and **Fig S5 in S2 Text**). The opposite is true for gene-wise distribution predictions, since model 1 is closer to the observations than later models (with the exception of the ΦX174+1 µg/ml gentamicin environment). This indicates that gene length is a reasonable

**Table 2. Chi-squared tests to check goodness of fit between observed and predicted mutant proportions.**

| LPS type (expected and observed proportions of deep rough and rough mutants, Fig 5A) | | | | |
|---|---|---|---|---|
| Treatment | Model 1 (added genes) | Model 2 | Model 3 | Model 1 |
| Phage | 0.0046 | 0.576 | 0.576 | 0.0005 |
| Phage + 1 µg/ml CL | 0.0368 | 6.35E-06 | 6.35E-06 | 0.0779 |
| Phage + 2 µg/ml CL | 0.0124 | 0.000388 | 0.01052 | 0.0124 |
| Phage + 1 µg/ml GM | 0.0084 | 0.429 | 0.429 | 0.0012 |
| Phage + 2 µg/ml GM | 0.7795 | 0.059 | 0.555 | 0.838 |
| **Gene-wise distribution (expected and observed proportions of mutations in each gene)** | | | | |
| Treatment | Model 1 (added genes)^ | Model 2 | Model 3 | |
| Phage | 6.20E-05* | 0.9981 | 0.9981 | |
| Phage + 1 µg/ml CL | 2.89E-06 | 5.16E-08 | 5.16E-08 | |
| Phage + 2 µg/ml CL | 0.0492 | 2.04E-06 | 1.00E-04 | |
| Phage + 1 µg/ml GM | 0.0185 | 0.0193 | 0.0193 | |
| Phage + 2 µg/ml GM | 0.0052 | 4.53E-06 | 5.20E-05 | |

*Values shown are *p*-values obtained by comparing the observed distribution of mutants (gene-wise or LPS type) with the distribution predicted by the given model.
^Before comparing model 1, 2 missing genes, *hldD* and *gmhA*, were added to genes conferring phage resistance to avoid zero expectation for any category. The data (observed mutations and predicted models) underlying this figure can be found in S4, S6, and S7 Tables and summarised in S5 Table.

proxy for the likelihood of observing a mutation in a particular gene (bottom of **Table 2**). Hence, adjusting the mutation proportions to the ΦX174-only environment leads to fewer errors for resistance rate and LPS type predictions, but greater errors for gene-wise predictions.

## Chloramphenicol addition leads to strongest synergism in co-evolving *E. coli* C-ΦX174 populations

To test whether our above predictions also hold in more complex environments, we set up a serial transfer co-evolution experiment. The experiment began with ten distinct *E. coli* C-ΦX174-antibiotic combinations (see **Fig 6A**). The growth of 12 replicate cultures in each environment was recorded for 24 h (day 1), after which 2% of each culture was transferred to fresh media (with the same antibiotic concentrations as day 1) and growth continued for a further 24 h (day 2). The most influential factor in the growth of day 1 cultures was the presence of phages; as expected, cultures with phages grew significantly more slowly than those without phages (**Fig 6B and 6C**, left panels). However, even cultures containing phages showed considerable bacterial growth by 24 h, suggesting the rapid emergence of phage-resistant bacteria (including the emergence of phage-resistant genotypes). Indeed, on day 2, the presence of phages is no longer the most influential factor on growth of the bacterial population (**Fig 6B and 6C**, right panels). Instead, the growth profiles of day 2 populations group not by phage presence, but instead by antibiotic: cultures with no antibiotic grew most quickly, then those with gentamicin, and finally those with chloramphenicol. Antibiotics becoming the dominating factor in our evolution experiment at day 2 is strong evidence of rapid phage resistance evolution. Further, the course of evolution is affected by the presence of antibiotics: phage resistance emerges rapidly on day 1 when no antibiotic is present, but considerably more slowly when chloramphenicol is present. In 2 µg/ml gentamicin, phage resistance emerges without a delay, yet bacterial growth is significantly inhibited. On day 2 for high antibiotic concentrations, there is no significant difference between the antibiotic only and the antibiotic-phage treatments except for the 1 µg/ml chloramphenicol environment.

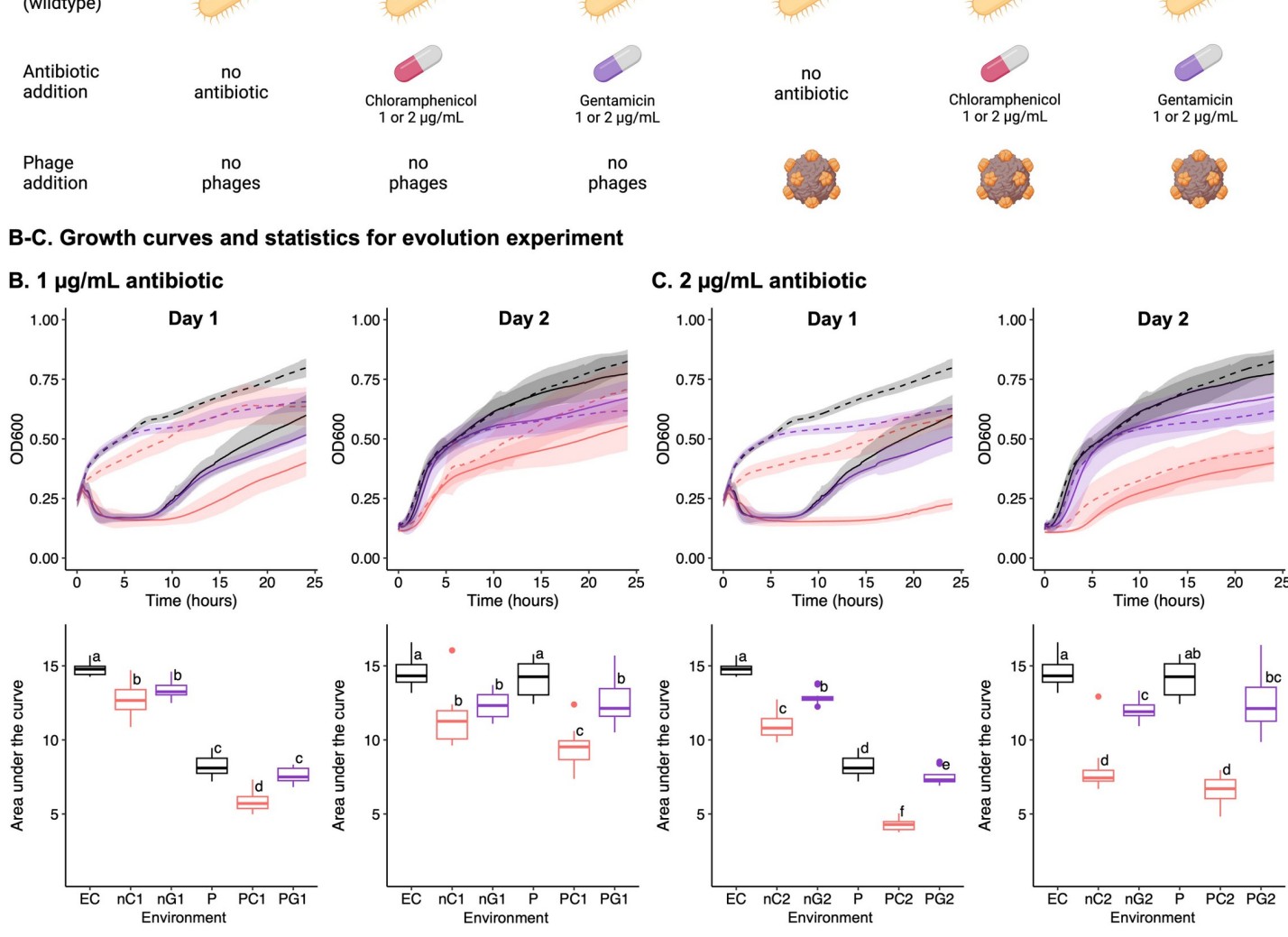

**Fig 6. Chloramphenicol affects co-evolution of *E. coli* C and ΦX174.** (**A**) Details of the 10 distinct *E. coli* C culture environments in the 2-day evolution experiment, containing the indicated combinations of ΦX174, chloramphenicol, and/or gentamicin (*n* = 12 per environment). Day 2 populations were founded by transferring 2% of each population into fresh media under the same conditions as day 1. (**B, C**) The effect of 1 μg/ml (**B**) or 2 μg/ml (**C**) antibiotic on the growth of *E. coli* C cultures with and without ΦX174. Growth is presented as absorbance measures over time (top) and area-under-the-curve (AUC, bottom), on day 1 (left), and day 2 (right). Letters on the AUC plots denote statistical significance groups for each day (a–f): there is no statistically significant difference between environments that share a particular letter on one day, while there is a significant difference between environments that do not share the letter. For example, on day 1, in 1 μg/ml antibiotic (**B**), there is no significant difference between nC1 and nG1, or between P and PG1, but all other pair-wise combinations are significantly different (Tukey's Honest Significant Difference test, see **Methods**). Statistical testing was done separately for (**B**) and (**C**). The data underlying this figure can be found in https://doi.org/10.17617/3.PIDVUT in the file Figure6_Platereader_raw_data.xlsx. Panel A created with BioRender.com.

Only for the 1 μg/ml chloramphenicol environment remain phages a significant growth inhibitor on day 2 (**Fig 6B**). This result is in line with the low level of phage resistance evolution at 1 μg/ml chloramphenicol in the solid fluctuation experiment (**Fig 3B**). Similarly, we posit that interactions between phages and antibiotics are likely the cause for the continued

efficacy of phage treatment and prevent the emergence of fast-growing phage-resistant mutants.

## Discussion

Here, we find that phage–antibiotic combinations can reduce the evolution of phage-resistance in bacteria, by inhibiting the growth of certain bacterial mutants. We start by quantifying the effect of chloramphenicol and gentamicin on the growth of 34 previously isolated ΦX174-resistant *E. coli* C strains. From these measurements, we predict that certain classes of mutations will not emerge in specified phage–antibiotic combination treatments, and hence these combination treatments will reduce the rate of phage-resistance evolution. We test these predictions by performing an experiment where we expose *E. coli* C wild type to phage ΦX174 and antibiotics. The observed phage resistance rates generally agree with our predictions and whole-genome sequencing of phage-resistant mutants support our predictions of mutational landscape accessibility. Overall, we show that antibiotics can reduce phage resistance evolution by preventing the growth of deep rough LPS resistance types. This effect is often referred to as an evolutionary trap, and can be useful for evolutionary steering [27,47,48]. Interestingly, first order effects—effects which do not take interactions between phage and antibiotic into account—are largely sufficient to explain reduced phage resistance evolution in combination treatments. Finally, we show that our predictions hold even in a more complex co-evolution experiment, where phage-chloramphenicol combinations are the most potent treatments for slowing bacterial growth.

### Phage–chloramphenicol interactions likely amplify the effect of combination treatments

The original model assumption for the fluctuation assay posits that the outcome of the assay only depends on 2 parameters: the mutation rate and the mutational target size [39]. The mutation rate should be identical across all treatments, since mutations occur during exponential growth in liquid culture in the absence of antibiotic and phages. The second parameter, the measured mutational target size, is very close to our expectations for 1 µg/ml chloramphenicol and similar to the measured target size for 2 µg/ml gentamicin (**Fig 4A**). Yet, for gentamicin the predicted phage resistance rate (61%) fits well with the observed resistance rate (60%), while for chloramphenicol the rates differ significantly (64% predicted versus 15% observed, **Fig 3B**). Given that target size and mutation rates are identical between the environments, there must be other factors that explain our observations.

One possible explanation is that mutations do not guarantee resistance. It is possible that for 1 µg/ml chloramphenicol the mutations only provide a certain survival chance rather than guaranteed survival. Previous studies have shown that while low concentrations of chloramphenicol do not lead to stochastic population extinctions, combination of chloramphenicol and a bactericidal antibiotic does [49]. Given that phages can be considered bactericidal agents, stochastic population extinctions could play a role in our experiment. If the stochastic extinction chance is around 50%, then the expected resistance rate would correspond to our observed resistance rate. This stochastic survival effect may result from several possible factors, including physiological interactions between phage and chloramphenicol on the plate, local fluctuations in chloramphenicol concentration, or differences in physiological cell state [50].

The combination between a bactericidal agent (phage) and a bacteriostatic antibiotic (chloramphenicol) may lead to a trade-off explaining why phage resistance rates at 1 µg/ml and 2 µg/ml chloramphenicol are similar. At high chloramphenicol concentrations, the antibiotic can efficiently prevent the emergence of deep rough mutants. However, the efficiency of

killing by the phage may be simultaneously compromised because of reduced bacterial growth. Slower growing bacteria inhibit phage infection and hence phage killing efficacy [49,51]. At slightly lower concentrations, chloramphenicol has a smaller effect on bacterial growth, rendering phage infection more efficient. These effects could conceivably balance each other out, leading to similar phage resistance rates at chloramphenicol concentrations of 1 µg/ml and 2 µg/ml.

Similar to our results in solid media, we also observe the strongest effect of combining phage and antibiotic in the 1 µg/ml chloramphenicol environment. This phage–antibiotic combination is the only environment where phages remain potent even on day 2 of the experiment. We propose that the interaction between antibiotic and phage prevent the evolution of phage resistance. The mechanism behind this interaction remains to be uncovered.

## Mutants resistant to phage-only treatment are biased towards deep rough LPS

In the phage-only environment, mutants with a deep rough LPS emerge at a higher proportion than expected based on the assumption that mutations occur in proportion to gene length. We observed that 86% of all mutants display the deep rough LPS phenotype (compared to the expected 58%, based on gene length) (**Fig 5B**). It is unclear why deep rough mutants are overrepresented, but one reason could be that mutation rates are not uniform throughout the genome. It is possible that there are many loci with high mutation rates in genes that cause deep rough phenotypes (mutational bias) [43,52,53]. Similarly, deep rough LPS candidate genes may contain a higher number of mutational targets (i.e., loci that produce a phage-resistant LPS phenotype) than do rough LPS candidate genes. Alternatively, mutations in deep rough LPS genes may be more likely to produce robust resistance—the ability of the mutation to reliably result in the establishment of a colony—in the presence of phages alone.

While deep rough LPS mutants are more abundant—at least in *E. coli*—they are more susceptible to certain antibiotics [21,54]. This characteristic could potentially be exploited by complementing phage therapy with antibiotics that specifically affect bacterial strains with a deep rough LPS structure.

## LPS modifications affect susceptibility to chloramphenicol and, to a lesser degree, gentamicin

Chloramphenicol and gentamicin are similar in their mechanism of action [33] and resistance mechanism [29,55–57]. Thus, the reason that chloramphenicol affects deep rough mutants differently may hinge on its physicochemical properties. The two antibiotics differ in their type of inhibition, hydrophobicity, and molecular size. Specifically, chloramphenicol is a bacteriostatic, hydrophobic molecule of 323 Da, while gentamicin is a bactericidal, hydrophilic molecule of 477 Da. The key differences are most likely hydrophobicity and size, because small hydrophobic molecules can diffuse through the LPS barrier, while hydrophilic molecules generally require active transport for cell entry [28].

The increased susceptibility of deep rough LPS mutants to hydrophobic antibiotics is in line with previous studies. Deleting deep rough LPS genes (*waaC*, *waaF*, *waaG*, or *waaP*) increases membrane hydrophobicity and permeability [54] can destabilise the membrane and increases the entry of small hydrophobic molecules [21,58,59]. Testing more antibiotics could reveal whether type of inhibition, molecular size, or hydrophobicity has the strongest effect on susceptibility.

## Host range extension experiments could make ΦX174 a potent therapeutic agent

Understanding the combined effect of phages and antibiotics on bacterial evolution ultimately aims to improve therapeutic outcomes. Unfortunately, ΦX174 itself is currently not a therapeutically relevant phage since it infects only about 0.8% of 783 *E. coli* human and animal commensal strains [22], and no pathogenic strains. However, the host range of ΦX174 can be expanded through evolution experiments [16,60] or genetic engineering [61]. Once the limitation of phage entry is overcome, ΦX174 can replicate its DNA and successfully lyse hosts, including various *E. coli* strains and other species such as *S. enterica* and *Klebsiella aerogenes* [62–64]. Extending the host range of ΦX174 could allow the extensive knowledge of ΦX174 biology accumulated over the last few decades to be used to convert ΦX174 into an efficient therapeutic agent, especially when combined with antibiotics to exploit shortened LPS structures [60].

More generally, our results show that phages used in combination with low levels of certain antibiotics can reduce the evolution of phage resistance in bacteria. This result indicates that the continued use of antibiotics may be beneficial even when the target bacteria are antibiotic-resistant. Of course, further studies are required to test whether our results also hold for higher, more clinically relevant antibiotic concentrations and resistance rates.

LPS modifications may also reduce pathogen virulence when treating with LPS-targeting phages. LPS can increase virulence [65–67] and elicit immune responses [68]. If the evolution of phage-resistance leads to the truncation of LPS structures, then developing resistance to phage infection may render the pathogen less virulent. Future experiments with clinically relevant antibiotics and phages should be conducted to allow us to understand which LPS-targeting phage–antibiotic combinations are most beneficial in therapeutic settings.

## Methods

### Bacteria and phage used in this study

*E. coli* C wild type and phage ΦX174 (GenBank accession number AF176034) were provided by H.A. Wichman (University of Idaho, United States of America). The *E. coli* C wild type used in this study differs from the NCBI *E. coli* C strain (GenBank accession number CP020543.1) by 9 insertions and 2 nucleotide replacements (see [16]). *E. coli* C phage-resistant strains R1 through R35 were generated in a previous study [16]; all other phage-resistant strains were generated in this study.

### Media and growth conditions

Bacteria and phages were grown in lysogeny broth (LB, Miller) supplemented with 5 mM $CaCl_2$ and 10 mM $MgCl_2$ to (henceforth referred to as "supplemented LB"). Supplemented LB agar (1.5%) was used to plate phages and bacteria. Plates were incubated for approximately 17 h at 37°C. Where appropriate, chloramphenicol or gentamicin was added at the concentration indicated. For bacterial overnight cultures, 5 ml supplemented LB in a 13 ml culture tube (Sarstedt) was inoculated with a single colony from a supplemented LB agar plate and incubated for approximately 17 h (37°C, 250 rpm). Fresh ΦX174 stocks were prepared weekly, using *E. coli* C wild-type cultures in exponential phase as described previously [16].

### Determination of antibiotic susceptibility of bacterial strains

**Determination of MIC.** Preliminary measurements of the antibiotic susceptibility of *E. coli* C R1 through R35 were obtained using ETEST strips (bioMérieux, France) on

supplemented LB agar plates, in order to narrow down the range of concentrations to be tested for MIC determination. Overnight cultures of *E. coli* C wild type and phage-resistant strains were diluted 10-fold in Ringer's solution; 50 μl of each diluted culture (approximately $5 \times 10^6$ cells) was spread, using approximately 15 glass beads (4 mm diameter), on a 90 mm supplemented LB agar plate (with or without antibiotic, as indicated). Plates were incubated at 37°C overnight (approximately 17 h). The next day, plates were cooled to room temperature and imaged for growth assessment. If bacterial growth was present on the images of all tested concentrations, a new round of testing with increased antibiotic concentrations was instigated. Tests were continued until there was no detectable growth across 3 replicates [69].

**Imaging to measure growth/susceptibility.** Supplemented LB agar plates were uniformly photographed using a Chemidoc imaging system (Bio-Rad) under the colorimetric setting with an exposure of 1 s (see **Data Availability**). Images were analysed using OMERO (web server, version 5.14.1) to quantify growth [70]. For uniform measurements, a rectangular region with a constant area (1,157.85 mm$^2$, an area size free of light glare in most images) was marked on each image. Growth was measured in 8-bit pixel greyscale values from 0 to 255. Growth measures were obtained by subtracting the average greyscale value of a blank plate from the average greyscale value of each sample plate (**S1 Table**). A uniform bacterial lawn corresponded to greyscale values of 30 through 80 (depending on the thickness and density of the lawn). No, or very little, visible growth corresponded with greyscale values of 0 through 8. Average greyscale values of more than 8 are defined as "growth," and values of less than 8 are defined as "no growth." The sensitivity of the assay to the growth cut-off was tested by changing the growth cut-off to 12 in model 3 (**S7 Table**). Median growth values across all (1 to 5) biological replicates were used for **Fig 2**. The lowest concentration at which the median value was less than the threshold (8 or 12) was considered the MIC.

## Fluctuation test and isolation of phage-resistant *E. coli* C mutants

Phage-resistance rates were measured using a revised Luria–Delbrück fluctuation test [39,40], across 9 plating environments (*n* = 50 per environment): (set 1) ΦX174 alone; (sets 2 to 4) ΦX174 in combination with chloramphenicol (at 1 μg/ml, 2 μg/ml, or 6 μg/ml); (sets 5 to 7) ΦX174 in combination with gentamicin (at 2 μg/ml, 2 μg/ml, or 4 μg/ml); (set 8) 6 μg/ml chloramphenicol alone (exception, *n* = 30); (set 9) 4 μg/ml gentamicin.

For each environment, 50 single *E. coli* C wild-type colonies picked from supplemented LB agar plates. Each colony was inoculated, in its entirety, into 100 μl of supplemented LB without antibiotic or phage (in 96-well plates). Inoculated cultures were incubated for 5 h (37°C, 750 rpm). Cultures were resuspended by pipetting at the 3-h and 4-h time points to avoid settling from bacterial aggregation. After incubation, 50 μl of culture from each well was spread onto the corresponding supplemented LB agar plate. For all environments involving ΦX174, bacterial cultures were mixed with 50 μl of a high-titre phage lysate (~$3 \times 10^8$ PFU/ml) prior to plating. Following initial incubation, plates with chloramphenicol at 6 μg/ml (with and without phage) were incubated further for 16 h to check for slow-growing colonies.

We designed the experiment to be able to accurately measure rates from $10^{-6}$ per cell division to $10^{-10}$ per cell division. Specifically, the amount of bacterial culture to be plated on selection plates was altered to have a countable number of resistant colonies [40,41]. Using a pilot experiment, we determined the size of bacterial culture to be plated for the expected number of colonies to be above 1 but below 100. For environment (set 1), 50 μl of 2-fold diluted bacterial culture was mixed with 50 μl of ΦX174 lysate; (sets 2 to 4) and (sets 5 to 7), 50 μl of undiluted bacterial culture (~$3 \times 10^8$ CFU/ml) was mixed with 50 μl of ΦX174 lysate; for (set 8) and (set 9), 50 μl of 10-fold diluted bacterial culture was taken. The entire mixture was then plated

on agar plates with the corresponding selective condition. For obtaining total CFU counts, cultures were serially diluted and plated on non-selective supplemented LB agar plates.

The next day, colonies were counted and 1 colony was randomly chosen from each replicate for which colonies were observed. Colonies were purified on non-selective supplemented LB agar plates, grown up in liquid culture, and stored at −80˚C. In total, 222 mutants were stored in this way (ΦX174-only $n = 50$; ΦX174+CL(1 μg/ml) $n = 41$; ΦX174+CL(2 μg/ml) $n = 37$; ΦX174+GM(1 μg/ml) $n = 46$; ΦX174+GM(2 μg/ml) $n = 43$; ΦX174+GM(4 μg/ml) $n = 1$; ΦX174+CL(6 μg/ml) $n = 4$).

### Resistance rate calculations and statistical analyses

Phage-resistance frequencies in *E. coli* C (i.e., the number of resistant colonies across 50 independent replicates) were used to calculate resistance rates. Resistance rates were calculated by the Lea–Coulson model following the method of Foster [71]. This method was used because (i) the bacterial populations were each founded by a single cell; (ii) the model assumes that the growth rates of mutants and non-mutants are the same in the absence of selection; and (iii) the model allows for partial plating of cultures. The resistance rate and 95% confidence interval were estimated using the Lea–Coulson ($\varepsilon<1.0$, i.e., partial plating) method of *webSalvador* (the web-application of R package *rSalvador*) [41]. Resistance rates were compared between different plating environments using the Lea–Coulson ($\varepsilon<1.0$) LRT from the R package *rSalvador* (RStudio version 2022.02.1) [72,73].

### Whole genome re-sequencing

Samples were prepared for whole genome re-sequencing from 500 μl of overnight culture, using a DNeasy Blood and Tissue kit (Qiagen). Samples were sequenced at the Max Planck Institute for Evolutionary Biology (Plön, Germany) with a NextSeq 500, using a NextSeq 500/550 High Output Kit v2.5 (300 Cycles) kit (Illumina). Output quality was controlled using *FastQC* version 0.11.9 [74], read-trimmed using *Trimmomatic* [75], and analysed using *breseq* [76]. Aside from SNPs, other predicted mutations were confirmed by mapping raw reads to the corresponding modified genome in Geneious Prime (2023.1.1, https://www.geneious.com).

### Predicting phage-resistance rates

**Model 1.** The probability of a ΦX174-resistance mutation occurring in a specific *E. coli* C gene is assumed to be proportional to the length of the gene's coding region (**Table 1**). We assumed that only genes previously observed [16] confer resistance to ΦX174. In cases where a phage-resistant *E. coli* C mutant carried multiple mutations, only the mutation predicted to affect LPS structure was considered (e.g., *waaO* for R29, *galE* for R1).

The observed ability of phage-resistant mutants to grow in the presence of chloramphenicol or gentamicin was used to predict (i) phage-resistance rates; and (ii) genes in which mutations can confer resistance in phage–antibiotic combination environments (**Fig 2** and values in **S1 Table**). If, out of "$n$" mutants with a mutation in a gene, only "$x$" mutants grew in the presence of the antibiotic, ($x/n$) was assumed to be the probability that this gene confers resistance in a combination environment. Hence, phage-resistance caused by a mutation in such a gene is proportional to $(x/n)^*$ gene length (i.e., a weighted score). For example, out of 5 *galU* mutants, only one can grow at 2 μg/ml chloramphenicol. Thus, resistance caused by a mutation in *galU* (909 bp) has a weighted score of 182 (1/5*909) in ΦX174+chloramphenicol (2 μg/ml).

Resistance rate is proportional to the sum of all weighted scores of genes conferring resistance. The relative resistance rate for a combination environment is inferred as a ratio of the

total weighted scores under each selective condition to that of the ΦX174-only environment (**Table 1**).

**Model 2.**   The weighted scores assigned to each gene were changed to the number of mutations actually observed in the ΦX174-only environment (**S6 Table**). Mutations in 2 genes (*gmhB* and *galE*) are not observed in the ΦX174-only environment, meaning a 0 frequency, and 0 weighted score, in model 2. A weight of 0 incorrectly classifies these genes as not conferring phage-resistance and will also predict 0 probabilities for combination environments. This issue was solved by adding a pseudo-count of 1 to all genes, including *galE* and *gmhB* (green highlight in **S6 Table**).

The inverse problem exists for *hldD* and *gmhA*. These genes were not observed in [16]; hence the degree of antibiotic susceptibility incurred through mutations in these genes was not measured in Fig **2**. Hence, phage-resistance rates and LPS type/gene distributions were calculated for both phenotypes. For final analyses, the model closest to the experimental observations (**Fig 2**) was used. *GmhA* mutants were assumed to grow similarly to *gmhB* mutant (i.e., growth in all environments except ΦX174+chloramphenicol (2 μg/ml)).

**Model 3.**   The growth cut-off greyscale value was increased from 8-bit to 12-bit (see **Methods**: section "Determination of antibiotic susceptibility of bacterial mutants"). Candidate genes and weighted scores of the baseline were the same as in model 2 (**S7 Table**).

Note: When comparing the 3 models with each other, model 1 was modified to include *hldD* and *gmhA* to avoid the statistical problem of zero expectation for any category (see "model 1 (added genes)" in **Table 2**).

## Bootstrap analysis

A bootstrap analysis was performed to estimate sampling error in the 217 *E. coli* C strains isolated from phage–antibiotic combination environments (only 1 or 2 μg/ml antibiotic, hence 5 mutants from other environments were excluded). Mutants were re-sampled with replacement 100 times for each environment, in order to calculate the expected variation in the sampled observation. To re-sample genes in which mutations were not observed, but are nonetheless expected to confer phage-resistance when mutated, a pseudo-count of 1 was added to each gene that could provide phage-resistance in that environment (before sampling). The standard deviation of the 100 sampling events was used to draw error bars for both gene-wise and LPS-type distributions (**Fig 5** and **Fig S5 in S2 Text**).

## Co-evolution experiment

**Environments.**   In a 2-day co-culture experiment, we measured how cultures of *E. coli* C (and its resistant mutants) grow when exposed to phage–antibiotic combinations, compared to the presence of either phages or antibiotics alone. Growth was measured across 10 environments: (sets 1 and 2) ΦX174 with chloramphenicol at 1 μg/ml or 2 μg/ml; (sets 3 and 4) ΦX174 with gentamicin at 1 μg/ml or 2 μg/ml; (set 5) ΦX174 only; (sets 6 and 7) chloramphenicol only, at 1 μg/ml or 2 μg/ml; (sets 8 and 9) gentamicin only, at 1 μg/ml or 2 μg/ml; (sets 10) no phage or antibiotic. We repeated the whole experiment twice and combined the data in the final plots, resulting in a total of 12 replicates for each environment (**Fig 6**).

**Day 1.**   Six independent overnight cultures of *E. coli* C wild type were used to inoculate 6 pre-cultures, by adding 100 μl of overnight culture to 5 ml supplemented LB. Once inoculated, the pre-cultures were incubated for 2 h (37°C, 250 rpm) with loosened caps, to obtain growing cells. Next, co-evolution experiment cultures were set up in 96-well plates as follows. To avoid evaporation issues, the 36 wells around the edges of the plate were not used, and filled with 150 μl supplemented LB to serve as blanks/media controls. To the remaining 60 wells, in a

randomised fashion (see **Data Availability**), a defined combination of bacteria, phage, and/or antibiotics were added (150 µl culture in total; see **Fig 6**) ($n$ = 6 per combination). To each well, 130 µl of either of the 6 bacterial pre-cultures was added. Phages were added at an approximate multiplicity of infection (MOI) of approximately 0.25 to 0.3 (~$3*10^7$ CFU of *E. coli* C and ~$7.5*10^6$ to $9*10^6$ PFU of ΦX174), and antibiotics were added at concentrations of 1 µg/ml or 2 µg/ml (**Fig 6A**). The plate was incubated in a microplate reader (Agilent BioTek Epoch 2 Microplate Spectrophotometer) for 24 h (37˚C, 365 rpm double orbital shaking). Growth was measured as absorbance at 600 nm (OD600) every 5 min.

**Day 2.** Approximately 3 µl (2%) of grown culture (from day 1) was transferred to 147 µl fresh supplemented LB in a new 96-well plate, with the same antibiotic conditions as before. Phages were not added separately (aside from the phages present in the 3 µl transfer). The plate was incubated under the same conditions for 24 h.

## Statistical testing

*T* tests and Wilcoxon rank sum tests were used to determine whether there is a statistically significant difference in the mean concentration of antibiotic at which the growth of rough and deep rough *E. coli* C mutants is impeded (**Fig 2B**). *T* tests were performed using the stats package of R (R version 4.2.0, RStudio version 2023.12.1) [72,73]. Wilcoxon rank sum tests, with an exact solution for tied data sets, were performed using the EDISON WMW calculator [77]. Chi-squared tests for the goodness-of-fit of models (**Table 2**) were performed using the stats package of R (R version 4.2.0, RStudio version 2023.12.1) [72,73]. For the same, observed distributions of mutations by gene and by LPS type were compared with the distributions predicted by each model. In the 2-day coevolution experiment, environments were compared using the Tukey's Honest Significant Difference (HSD) test combined with ANOVA (**Fig 6B and 6C**). An ANOVA model was fitted to the area under the bacterial growth curves in each environment, followed by pairwise comparisons of the different environments using the Tukey's HSD test, both using the stats package of R (R version 4.2.0, RStudio version 2023.12.1) [72,73]. Growth on the different days was compared separately.

## Data visualisation

Heatmaps (**Fig 2**) and histograms (**Figs 3B, 3C**, **4B**, **5**, and **Figs S2 and S3 in S2 Text**) were generated using Microsoft Excel (version 16.75.2). Schematic drawings (**Figs 1**, **2,** and **3A**) were created using BioRender.com. The genomic distribution of mutation loci (**Fig 4A**) and boxplot of antibiotic susceptibilities (**Fig 2B**) were visualised using R (RStudio version 2022.02.1) [72,73].

## Supporting information

**S1 Text. Note A. Explanation of differences in antibiotic susceptibility profiles between isogenic bacterial strains in Fig 2. Note B. Parallel evolution in the fluctuation test. Note C. Predicted mutations from the fluctuation test that occur in genes not known to be related to LPS biosynthesis, assembly, or regulation.**
(DOCX)

**S2 Text. Fig S1. Raw mean greyscale values measured for strains grown in the presence of chloramphenicol.** The figure shows mean greyscale values for growth of different *E. coli* C strains combined with different chloramphenicol concentrations. There are between 1 and 5 measurements for each combination shown on the x-axis. Each measurement is the mean greyscale value across a region of 1,157.85 mm$^2$ in a photo of an LB plate inoculated with an *E.*

*coli* C strain (see **Methods**). The black horizontal line at a greyscale value of 37 indicates the mode of the mean background greyscale value of blank LB plates without addition of bacteria. The green horizontal line shows the strict growth cut-off of 8 (37+8) and the red horizontal line shows the lenient growth cut-off of 12 (37+12). The colours of the datapoints indicate whether the strain is predicted to possess a rough or a deep rough LPS phenotype. The data underlying this figure (greyscale values of replicates and raw images) can be found in https://doi.org/10.17617/3.PIDVUT. **Fig S2. Raw mean greyscale values measured for strains grown in the presence of gentamicin.** For details, see S1 Fig. **Fig S3. Mean greyscale values of strains grown in the presence of chloramphenicol (A) or gentamicin (B), clustered by whether the strain is rough or deep rough.** The figure shows the mean greyscale values for the growth of *E. coli* C at different antibiotic concentrations on LB plates for all strains shown in **Fig 2**. Each measurement is the mean greyscale value across a region of 1,157.85 mm² in a photo of an LB plate inoculated with an *E. coli* C strain. The black horizontal line at a greyscale value of 37 indicates the mode of the mean background greyscale value, that is, of blank LB plates without addition of bacteria. The green horizontal line shows the strict growth cut-off of 8 (37+8) and the red horizontal line shows the lenient growth cut-off of 12 (37+12). The colours of the datapoints indicate whether the strain is predicted to possess a rough or a deep rough LPS phenotype. The data underlying this figure (greyscale values of replicates and raw images) can be found in https://doi.org/10.17617/3.PIDVUT. **Fig S4. Types of mutations identified across phage-resistant *E. coli* C strains isolated from the fluctuation test.** Bars show all predicted mutations grouped by mutational class (as % of total mutations "*n*" identified in each of the 5 plating environments sampled). If an isolate had 2 mutations, they were classified and counted separately. Insertions and deletions are further separated into 4 categories: (i) transposon insertion, where the inserted bases are a transposable element, (ii) duplication, where the inserted bases match the sequence preceding the mutated locus (up to 1.3 kbp), (iii) large deletion, where a region of more than 100 bp is deleted, and (iv) small indels, other insertions and deletions of <100 bp. CL = chloramphenicol, GM = gentamicin. The data underlying this figure (mutations) can be found in **S4 Table**. **Fig S5. Comparison of model 3 predictions with observed phage-resistance rates.** (**A**) Phage-resistance rates for phage–antibiotic combinations, relative to the phage-resistance rate in the ΦX174-only environment. Observed rates (black) are only qualitatively in line with the predicted rates from model 1 (gene length model, dark red; see **Table 1**). Rates predicted by model 2 (dark yellow; derived in **S6 Table**) are within 95% confidence intervals for ΦX174+gentamicin environments and are 2% away from the 95% confidence interval in ΦX174+chloramphenicol (2 µg/ml) environment. Phage-resistance rates for model 3 (blue, **S7 Table**) remain largely unchanged (relative to those for model 2), except for the ΦX174+gentamicin (2 µg/ml) environment (where the prediction becomes worse). (**B**) LPS phenotype distribution in phage-resistant *E. coli* C mutants isolated in the fluctuation test. Graphs show proportion of mutants with a certain LPS phenotype (values in **S5 Table**). Model 1 predictions (dark red) are qualitatively in line with predictions. In model 2 (dark yellow), LPS phenotype distribution in the ΦX174-only environment is now in line with the observations. Model 3 predictions (blue) improve slightly for ΦX174+chloramphenicol (2 µg/ml) and ΦX174+gentamicin (2 µg/ml) environments. (**C–F**) Gene-wise distribution compared to model 3 predictions (and model 2 in C and D). In ΦX174 +chloramphenicol (1 µg/ml), mutations are predicted in *waaF*, *gmhB*, *gmhA*, and *rfaH*, but none are observed. Similarly, in ΦX174+gentamicin (2 µg/ml), mutations in *gmhB*, *gmhA*, *hldE* are predicted but not observed. Compared to model 2, only predictions at ΦX174+-chloramphenicol (2 µg/ml) (*waaP*, *galU*) and ΦX174+gentamicin (2 µg/ml) (*gmhB*, *gmhA*, *hldE*) change in model 3. Error bars show the standard deviation for each LPS phenotype (**B**) or each gene (**C–F**) from 100 bootstrapping iterations (resampling with replacement, see

**Methods**). The data underlying this figure can be found in **S4–S7 Tables**, which include raw mutation data as well as summarised values.
(DOCX)

**S1 Table. Growth of *E. coli* C strains in chloramphenicol and gentamicin.** *No antibiotic: Growth on supplemented LB plates without antibiotic. #Median values of the mean pixel density in bacterial lawns, after subtracting mean background density (see **Methods**). Source data for **Fig 2**. Raw replicate values are plotted in **Figs S1–S3 in S2 Text**. The data underlying this figure (images of LB plates) can be found in https://doi.org/10.17617/3.PIDVUT.
(XLSX)

**S2 Table. Resistance rates observed in the fluctuation test.** **Nt is the total number of cells in the culture. #ε is the fraction of cells plated. $We counted phage-resistant colonies from 50 biological replicates for each plating environment, which were used to calculate phage-resistance rates and 95% confidence intervals (with rSalvador; see **Methods**). $The likelihood-ratio test statistic from rSalvador was used to determine whether each calculated phage-resistance rate is significantly different to that of the ΦX174-only plating environment. *Mutation rates and significance intervals for GM 4 μg/ml were calculated without counting plates that contained too many colonies. †Resistance rate relative to the resistance rate in the ΦX17-only plating environment, calculated for combination plating environments with chloramphenicol (CL) or gentamicin (GM). Raw data (colony counts) are provided in **S3 Table**. Source data for **Fig 3**.
(XLSX)

**S3 Table. Colony counts from the fluctuation test.** **Phage-resistant colonies were counted from 50 biological replicates for each fluctuation test plating environment. The resulting colony counts were used to calculate phage-resistance rates and 95% confidence intervals (see **Methods**). *"Burst": some plates contained areas with large numbers of tiny colonies (i.e., uncountable). #"invalid": Plate discarded due to contamination. Source data for **Fig 3**.
(XLSX)

**S4 Table. Details of mutations predicted in the phage-resistant *E. coli* C strains isolated from the fluctuation test.** $DUP indicates a duplication event. #"2,611,543, +A": indicates an insertion of "A" after nucleotide 2,611,543. **"2,611,295–2,611,303, INS (2,783,815–2,784,945 INV)": indicates positions 2,783,815–2,784,945 were inserted (inverted) at position 2,611,303. The inserted nucleotides are flanked by 2,611,295–2,611,303 on both sides (i.e., 2,611,295–2,611,303 are duplicated). $E187fs_Ter#192 indicates a frameshift starting at E187, with a stop codon (Ter) generated at codon #192. F366mod_Ter#383 indicates protein sequence modified, e.g., due to a large insertion/duplication starting from F366, and (premature) stop codon (Ter) generated at codon #383. Difference between frameshift (fs) and modification (mod) is whether the original sequence of the gene is used as coding region with only a frameshift, or the new inserted sequence is used until the stop codon. ‖Predicted LPS phenotypes for the mutants and the mutations in their genome. *Where multiple LPS genes carry mutations, the star indicates the gene predicted to cause the most dramatic LPS phenotype change. ^Column indicates whether the mutated gene is related to LPS biosynthesis, assembly, or regulation. †Even though the amino acid is unchanged, the codon affected is the start codon ATG, hence the initiation of translation is expected to be inhibited. See **Fig S4 in S2 Text** for a summary of mutational classes.
(XLSX)

**S5 Table. Observed and predicted distribution of mutations in LPS genes.** $Mutations in *rfaH* or promoter regions of the *waa* operon can affect any number of inner or outer core

genes, hence are in a separate category (*rfaH*\*). Source data for **Fig 5**.
(XLSX)

**S6 Table. Details of Model 2 used to predict phage-resistance rates.** See **Methods** for model 2 assumptions. \*\*New genes. Highlighted in green: 2 genes in which 0 mutations were detected among isolates from the ΦX17-only plating environment, due to which a pseudo-count or pseudo-weight of 1 was added to all genes (including these 2 genes).
(XLSX)

**S7 Table. Details of Model 3 used to predict phage-resistance rates.** See **Methods** for model 3 assumptions.
(XLSX)

# Acknowledgments

The authors thank the Barrick Lab for their fluctuation test protocols, Prof. Qi Zheng for his help with rSalvador, Dr. Carsten Fortmann-Grote for his help with image analysis, and all members of the Microbial Population Biology department who helped with experiments, especially Daniel Martens, Dr. Elisa Brambilla, and Lena Zeller.

# Author Contributions

**Conceptualization:** Lavisha Parab, Frederic Bertels.

**Data curation:** Lavisha Parab, Jenna Gallie, Frederic Bertels.

**Formal analysis:** Lavisha Parab, Frederic Bertels.

**Funding acquisition:** Lavisha Parab, Frederic Bertels.

**Investigation:** Lavisha Parab, Jordan Romeyer Dherbey, Norma Rivera, Michael Schwarz.

**Methodology:** Lavisha Parab, Jordan Romeyer Dherbey, Frederic Bertels.

**Software:** Lavisha Parab.

**Supervision:** Lavisha Parab, Jenna Gallie, Frederic Bertels.

**Validation:** Lavisha Parab, Jenna Gallie, Frederic Bertels.

**Visualization:** Lavisha Parab, Jenna Gallie, Frederic Bertels.

**Writing – original draft:** Lavisha Parab, Frederic Bertels.

**Writing – review & editing:** Lavisha Parab, Jordan Romeyer Dherbey, Jenna Gallie, Frederic Bertels.

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
