## [Editor Report · Decision Letter 0]

3 Oct 2024

Dear Dr Parab, 

Thank you for submitting your manuscript entitled "Chloramphenicol and gentamicin reduce resistance evolution to phage ΦX174 by suppressing a subset of E. coli C LPS mutants" for consideration as a Research Article by PLOS Biology.

The revised version of your manuscript has now been evaluated by the PLOS Biology editorial staff, as well as by the previous academic editor and I am writing to let you know that we would like to send your submission back to 2 of the previous reviewers.

Once your full submission is complete, your paper will undergo a series of checks in preparation for peer review. After your manuscript has passed the checks it will be sent out for review. To provide the metadata for your submission, please Login to Editorial Manager (https://www.editorialmanager.com/pbiology) within two working days, i.e. by Oct 05 2024 11:59PM.

Kind regards,

Melissa

Melissa Vazquez Hernandez, Ph.D.

Associate Editor

PLOS Biology

---

## [Decision Letter · Decision Letter 1]

31 Oct 2024

Dear Dr Parab,

Thank you for your patience while we considered your revised manuscript "Chloramphenicol and gentamicin reduce resistance evolution to phage ΦX174 by suppressing a subset of E. coli C LPS mutants" for publication as a Research Article at PLOS Biology. This revised version of your manuscript has been evaluated by the PLOS Biology editors, the Academic Editor and the original reviewers; being reviewer 1 Fernando Gordillo Altamirano.

Based on the reviews, we are likely to accept this manuscript for publication, provided you satisfactorily address the remaining points raised by the reviewers. Please also make sure to address the following data and other policy-related requests.

a) We routinely suggest changes to titles to ensure maximum accessibility for a broad, non-specialist readership, and to ensure they reflect the contents of the paper. In this case, we would suggest a minor edit to the title, as follows. Please ensure you change both the manuscript file and the online submission system, as they need to match for final acceptance:

"Chloramphenicol and gentamicin reduce the evolution of resistance to phage ΦX174 by suppressing a subset of E. coli LPS mutants"

b) Please provide the grant number in your financial statement.

c) We do not have a count limit. Please include the supplementary notes and references in the main text.

Please supply the numerical values either in the a supplementary file or as a permanent DOI’d deposition for the following figures and tables:

Figure 2AB, 3BC, 4AB, 5ABC, 6BC , S1, S2, S3AB, S4, S5A-E, and Tables 2, S1

e) Please cite the location of the data clearly in all relevant main and supplementary Figure legends, e.g. “The data underlying this Figure can be found in S1 Data” or “The data underlying this Figure can be found in https://doi.org/10.5281/zenodo.XXXXX”

f) Please ensure that your Data Statement in the submission system accurately describes where your data can be found and is in final format, as it will be published as written there.

g) Per journal policy, if you have generated any custom code during the course of this investigation, please make it available without restrictions upon publication. Please ensure that the code is sufficiently well documented and reusable, and that your Data Statement in the Editorial Manager submission system accurately describes where your code can be found.

We expect to receive your revised manuscript within two weeks. 

*Published Peer Review History*

*Press*

Sincerely,

Melissa

Melissa Vazquez Hernandez, Ph.D.

Associate Editor

PLOS Biology

REVIEWERS' COMMENTS

I congratulate the authors on their clear and substantive effort put into revising the manuscript following the previous rounds of comments. I think that this latest version tells a cohesive and compelling story that is nevertheless nuanced and easy to follow. I only have a few very minor comments:

Line 31: I recommend replacing "outer membrane" with "surface". Many phages use non-outer membrane receptors such as capsule or appendages.

Line 117: Replace "surprisingly" with "surprising"

Line 411, Table 2: Check the highlighting of cells in this table. I think there's a mistake in the gene-wise distribution: it's the cells for "Phage" in Models 2 and 3 that should be highlighted instead of the cells for "Phage + 1 ug/ml GM"

Line 426: I suggest you specify here, as you have done in the Methods section, that the "fresh media" in this transfer contained the previously-stated antibiotics as well.

Lines 438-439: The statement "but considerably more slowly when gentamicin or chloramphenicol is present" seems inaccurate. Looking at the figure, it seems that this is only true for cloramphenicol but not for gentamicin.

Lines 552-553: I suggest toning down the argument that, because of its narrow host range, ΦX174 is unsuitable for therapeutic purposes. A lot of therapeutic phages have narrow host ranges and have been successfully used in bespoke cases. Perhaps a better description is that ΦX174 is unsuitable for "widespread" therapeutic use?

Line 557: Enterobacter aerogenes is a misnomer. The species has been renamed as Klebsiella aerogenes, check https://www.ncbi.nlm.nih.gov/pmc/articles/PMC7448666/

Line 639: There is a typo duplicating the references by Zheng and Rosche and Foster.

Reviewer #2: 

I have reviewed two previous versions of this manuscript. The authors have now added some experimental data in liquid culture, that add to the fluctuation assays by allowing to observe some population dynamics in the presence of combination phage+antibiotic treatment, which supports their previous claims. They have also re-analysed their data on MIC effects of phage resistance mutants, with a clearer interpretation provided in Fig 2B. More generally, the manuscript has been re-written and I believe it reads much better now, with a clearer back and forth discussion between the model and experimental results. I'm happy to recommend it for publication, with only a few remaining comments: 

Minor comments

L55 - references for the claim of treatment failure due to phage resistance? 

L550-562 - I don't think host range considerations are relevant to this paper

---

## [Editor Report · Decision Letter 2]

25 Nov 2024

Dear Lavisha,

Thank you for the submission of your revised Research Article "Chloramphenicol and gentamicin reduce the evolution of resistance to phage ΦX174 by suppressing a subset of E. coli LPS mutants" for publication in PLOS Biology. On behalf of my colleagues and the Academic Editor, Jeremy Barr, I am pleased to say that we can in principle accept your manuscript for publication, provided you address any remaining formatting and reporting issues. These will be detailed in an email you should receive within 2-3 business days from our colleagues in the journal operations team; no action is required from you until then. Please note that we will not be able to formally accept your manuscript and schedule it for publication until you have completed any requested changes.

PRESS

Sincerely, 

Melissa

Melissa Vazquez Hernandez, Ph.D., Ph.D.

Associate Editor

PLOS Biology
